# Rationalizing Graphene–ZnO Composites for Gas Sensing via Functionalization with Amines

**DOI:** 10.3390/nano14090735

**Published:** 2024-04-23

**Authors:** Maxim K. Rabchinskii, Victor V. Sysoev, Maria Brzhezinskaya, Maksim A. Solomatin, Vladimir S. Gabrelian, Demid A. Kirilenko, Dina Yu. Stolyarova, Sviatoslav D. Saveliev, Alexander V. Shvidchenko, Polina D. Cherviakova, Alexey S. Varezhnikov, Sergey I. Pavlov, Sergei A. Ryzhkov, Boris G. Khalturin, Nikita D. Prasolov, Pavel N. Brunkov

**Affiliations:** 1Ioffe Institute, Politekhnicheskaya St. 26, Saint Petersburg 194021, Russia; gabri_fti@mail.ru (V.S.G.); zumsisai@gmail.com (D.A.K.); sviatoslav.saveliev@gmail.com (S.D.S.); alexshvidchenko@mail.ru (A.V.S.); pchervyakova@mail.ioffe.ru (P.D.C.); pavlov_sergey@mail.ioffe.ru (S.I.P.); ryzhkov@mail.ioffe.ru (S.A.R.); khalturin.boris@gmail.com (B.G.K.); nikpras@mail.ioffe.ru (N.D.P.); brunkov@mail.ioffe.ru (P.N.B.); 2Department of Physics, Yuri Gagarin State Technical University of Saratov, 77 Polytechnicheskaya St., Saratov 410054, Russia; vsysoev@sstu.ru (V.V.S.); solomatin1994@gmail.com (M.A.S.); alexspb88@mail.ru (A.S.V.); 3Helmholtz-Zentrum Berlin für Materialien und Energie, Hahn-Meitner-Platz 1, 14109 Berlin, Germany; maria.brzhezinskaya@helmholtz-berlin.de; 4NRC “Kurchatov Institute”, Akademika Kurchatova pl. 1, Moscow 123182, Russia; stolyarova.d@gmail.com

**Keywords:** two-dimensional (2D) material, functionalized graphene, graphene–metal oxide composite, e-nose, multisensor array, gas sensor, alcohol, ammonia

## Abstract

The rational design of composites based on graphene/metal oxides is one of the pillars for advancing their application in various practical fields, particularly gas sensing. In this study, a uniform distribution of ZnO nanoparticles (NPs) through the graphene layer was achieved, taking advantage of amine functionalization. The beneficial effect of amine groups on the arrangement of ZnO NPs and the efficiency of their immobilization was revealed by core-level spectroscopy, pointing out strong ionic bonding between the aminated graphene (AmG) and ZnO. The stability of the resulting Am-ZnO nanocomposite was confirmed by demonstrating that its morphology remains unchanged even after prolonged heating up to 350 °C, as observed by electron microscopy. On-chip multisensor arrays composed of both AmG and Am-ZnO were fabricated and thoroughly tested, showing almost tenfold enhancement of the chemiresistive response upon decorating the AmG layer with ZnO nanoparticles, due to the formation of p-n heterojunctions. Operating at room temperature, the fabricated multisensor chips exhibited high robustness and a detection limit of 3.6 ppm and 5.1 ppm for ammonia and ethanol, respectively. Precise identification of the studied analytes was achieved by employing the pattern recognition technique based on linear discriminant analysis to process the acquired multisensor response.

## 1. Introduction

Over the last decade, high expectations have been associated with combining 2D materials and metal oxide micro- and nanoparticles (NPs) for advancing performance in various practical applications, including catalysis, energy storage, and especially gas sensing [1,2,3]. The introduction of 2D materials, primarily graphene, is considered to represent a technology platform for the R&D of next-generation sensing systems, including on-chip multisensor arrays for so-called e-nose systems [4,5]. Two major strategies to form composites—(i) with metal oxide microparticles covered with graphene on the one hand, and (ii) graphene layers decorated by metal oxide NPs (Gr-MO) on the other—have been promptly developed to achieve the proper combination of gas-sensing properties, high response, and selectivity to analytes [3,6,7,8]. Despite being debatable, the latter approach is regarded as a more beneficial one, since it allows researchers not only to fully capitalize on the unique features of graphene—such as exceptional specific surface area, tunable electronic structure, and low levels of electrical noise [9,10]—but also to tackle the key drawbacks of metal oxide sensing units, such as their high operating temperature.

A large number of gas-sensing units with built-in multisensor e-nose devices based on Gr-MO gas-sensitive layers have been developed to date, employing SnO_2_, CeO_2_, and ZnO NPs as the most favored metal oxide representatives [6,8,11,12,13]. Among the mentioned MO NPs, ZnO ones are the most renowned, owing to their wide bandgap (3.37 eV), significant excitation binding energy (60 meV), and high electron mobility of ca. 400 cm^2^∙V^−1^∙s^−1^ [14]. Moreover, ZnO has the advantages of biocompatibility, chemical stability, environmental friendliness, and low synthesis cost [15,16]. Exhibiting a decent sensitivity level while operating at room temperature or slightly higher, composites based on graphene and other 2D materials bearing ZnO NPs showing advantages in the fabrication of miniaturized, robust, and low-power gas-sensing instrumentation for various practical applications, particularly for dealing with Internet-of-Things tasks [17].

Despite this progress, further advancement in the application of Gr-MO in gas sensing faces the problem of dealing with the fundamental issue of achieving homogeneous distribution of MO nanoparticles through the graphene layer, both during the formation of the composite and upon its further application [18]. Being governed by surface states of MO NPs as well as the total number of p-n local heterojunctions appearing in the areas of contact between NPs and the graphene layer [18,19], the gas-sensing properties of Gr-MO are heavily influenced by the arrangement of MO NPs at the graphene layer, achieving their maxima only upon uniform allocation. At the same time, this is barely feasible in the case of pristine graphene due to weak Van der Waals interactions.

To overcome this issue, functionalized graphenes can be employed [20,21], benefiting from the functional groups that serve as anchoring points to immobilize MO NPs and control their distribution efficiently. Graphene oxide is one of the most renowned functionalized graphenes, yet it is hardly suitable as an efficient platform for Gr-MO composites due to its low conductivity and chemical instability. In turn, aminated graphene (AmG) can be regarded as a material of choice for this purpose. AmG is presented by a pristine graphene layer, in which local areas are covalently functionalized by primary amines (-NH_2_) at concentrations of 3–10 at.% [22,23,24], with negligible contents of other organic groups. This distinguishes AmG from graphene oxide layers covered with amine-containing oligomers [25,26], which are often misidentified as aminated graphenes as well. Exhibiting extended areas of the π-conjugated graphene network on the one hand and a considerable number of chemically reactive electron-donating amine groups clustered in arrays on the other [23], AmG combines a conductive nature, tunable work function, modified band structure and, first and foremost, enhanced chemical activity towards covalent and electrostatic bonding. Thanks to the latter factor, AmG facilitates the coordination of MO NPs with their ordered distribution, while its tunable electronic structure allows one to tailor the gas-sensing properties. However, to the best of our knowledge, no reports on the synthesis of composites based on AmG covered with MOs have been published yet, due to challenges in the synthesis of AmG layers.

Herein, we report for the first time on the fabrication of the AmG-ZnO composite (Am-ZnO), achieving a uniform distribution and highly stable immobilization of NPs over the graphene layer thanks to its amination. The beneficial effect of amine groups is confirmed by means of core-level spectroscopy methods, namely, X-ray photoelectron spectroscopy (XPS) and X-ray absorption spectroscopy (XAS), which clarify the features of bonding between the AmG and ZnO NPs. On-chip multisensor arrays composed of both AmG and Am-ZnO were fabricated and thoroughly tested, signifying both the gas-sensing properties of AmG and their alteration upon the introduction of ZnO nanoparticles. The acquired results, on the one hand, advance the development and fabrication of graphene-based e-nose units for volatile organic compound (VOC) fingerprinting, while on the other hand they allow one to step forward in the field of designing Gr-MO composites, not only for gas sensing but also for other practical applications, such as catalysis and energy harvesting.

## 2. Materials and Methods

### 2.1. Materials

The aqueous dispersion of the initial GO employed for the synthesis of the aminated graphene was purchased from Graphene Technologies (Moscow, Russia, www.graphtechrus.com, accessed on 10 January 2024). Formamide (CH_3_NO) and ZnO NPs (product no. 793361, concentration 2.5 wt.%) were supplied by Merck KGaA (Darmstadt, Germany).

All of the organic solvents used in this work, as well as the ethanol and ammonia test analytes, were acquired from Vecton Ltd. (Saint Petersburg, Russia).

All of the chemicals were of analytical-grade purity, commercially available, and used as received without additional purification.

### 2.2. Synthesis of AmG and Am-ZnO Composite

To synthesize AmG, liquid-phase reductive amination of the initial GO was performed by the modified Leuckart reaction via the protocol given in [23]. Briefly, 100 mL of the GO aqueous suspension (0.3 wt.% concentration) was mixed with 50 mL of CH_3_NO in a Teflon reaction vessel, with subsequent heating up to 165 °C for 48 h while stirring. Afterwards, the reaction mixture was cooled down to room temperature (RT), placed in a glass filter (16 μm pore size), and the synthesized AmG was purified by rinsing it with water (3 cycles) and isopropyl alcohol (3 cycles) until the solution became colorless. Then, the acquired AmG powder was collected from the glass filter to prepare its dispersion in isopropyl alcohol (0.1 wt.% concentration), with the help of sonication (110 W, 35 Hz) for 5 min.

To prepare pristine rGO, a GO suspension (0.1 wt.% concentration) was dropped over a polyethylene terephthalate substrate to be further dried overnight at RT. The as-prepared GO paper was peeled off the substrate and ground in a jasper mortar, with a final annealing of the powder at 650 °C in an ultra-high-vacuum chamber at 10^−9^ Torr pressure for 3 h. For further use, the obtained rGO powder was dispersed in isopropyl alcohol, similarly to the AmG powder.

Am-ZnO and rGO-ZnO composites were fabricated by dropwise dispensing of AmG/rGO and ZnO suspensions; the latter was diluted down to a concentration of 0.01 wt.% by adding 250 mL of deionized water in the Teflon vessel, before being rigorously stirred by a magnetic mixer at 400 rpm. In total, 100 mL of AmG/rGO suspension and 88 mL of ZnO suspension were mixed to ensure a homogeneous distribution of ZnO NPs over the AmG/rGO layers.

To fabricate samples of the graphene derivatives under study for the spectroscopic and microscopic inspections, the corresponding suspensions (0.01 wt.% concentration) were drop-cast over the Si wafers or transmission electron microscopy (TEM) Cu grids (400 Mesh) and dried overnight at RT.

To examine the thermal stability of the Am-ZnO and rGO-ZnO structures, the samples were annealed at 315 °C for 3 h in a high-vacuum chamber at 10^−3^ Torr pressure.

### 2.3. Fabrication of Graphene-Based on-Chip Multisensor Arrays

AmG and Am-ZnO layers were placed over the on-chip multisensor arrays by spray coating according to protocols described elsewhere [27,28]. Both chips were cut from Si/SiO_2_ substrate at 9 × 10 mm^2^ in size, to be metalized with an array of 39 co-planar strip Pt (in the case of AmG) or Au (in the case of Am-ZnO) electrodes (50 μm wide, with 50 μm gaps) via magnetron sputtering (Emitech K575X, Ashford, Kent, UK) and photolithography patterning. The graphene-based materials under study were deposited over the chip by aerosol spraying at 125 mm distance in a home-made setup; AmG and Am-ZnO isopropyl suspensions (0.3 wt.% concentration) were applied. Filtered dry air was utilized as a carrier gas at a pressure of 2280 Torr, with a flow rate of 0.8 L/min. In total, 5 mL of AmG and 7 mL of Am-ZnO were deposited over a single chip. To localize the deposition of the graphene-based materials within the area of the on-chip electrodes, a dark-field mask at a rectangular aperture of 5 × 5 mm^2^ was applied. Following the deposition, the chip was additionally heated at 150 °C in the vacuum chamber, with 10^−2^ Torr pressure, for 4 h to remove any residuals and to stabilize the material properties.

### 2.4. Materials’ Characterization

A set of microscopic studies were carried out to examine the morphology of the AmG prior to and after the ZnO NPs’ deposition. Scanning electron microscopy (SEM) was carried out for the AmG and Am-ZnO films on Si wafers with the use of a JSM-7001F microscope (Jeol, Ltd., Tokyo, Japan). TEM images and electron diffraction (ED) patterns for the individual AmG and Am-ZnO platelets were collected with a Jeol JEM-2100F microscope (Jeol, Ltd., Tokyo, Japan).

The core-level spectroscopy was used to study the chemical composition of the materials under study, namely, XPS and XAS; this was carried out at the ultra-high-vacuum experimental station of the Russian–German beamline of electron storage ring BESSY-II at Helmholtz-Zentrum Berlin (HZB) [29]. For the measurements, all of the samples were evacuated down to a pressure of P~10^−10^ Torr for 24 h at RT to remove all adsorbates. To check the uniformity of the samples and acquire statistically relevant data, both XPS and XAS spectra were collected from four equidistant areas of the sample separated by ca. 500 µm, each of ca. 300 × 500 µm size. For both AmG and Am-ZnO, the differences between the spectra acquired from all the spots did not exceed 5%, signifying the uniformity of these graphene derivatives in terms of their chemical composition. Averaged spectra were further employed to perform the quantitative analysis.

The survey spectra, along with the Zn 2*p* core-level spectrum, were collected using excitation energy of 1200 eV, whereas all other core-level spectra (O 1*s*, N 1*s*, and C 1*s*) were collected with a lower excitation energy of 850 eV, to ensure higher photon flux intensity and, thus, a higher signal-to-noise ratio (*SNR*). For the survey spectra, an energy step of 0.5 eV and pass energy of 50 eV were chosen, while the core-level spectra were recorded with values of 0.05 eV and 20 eV, respectively. To estimate the elemental composition of the AmG and Am-ZnO, the survey spectra were processed, considering the relative sensitivity factors to be C 1*s* = 1, N 1*s* = 1.80, O 1*s* = 2.93, and Zn 2p_(3/2)_ = 18.92. CasaXPS@ software (version 2.3.16 Dev52, Casa Software Ltd.; Teignmouth, UK) was applied to deconvolute the collected core-level spectra. A Shirley background fitted all of the spectra, whereas different sets of functions were employed to display the spectral components. The C 1*s* spectra were deconvoluted with the help of a set of one asymmetric Doniach–Sunjic function (DS; 0.09–0.15; 90–250; GL90) and six symmetric Gaussian−Lorentzian functions with a ratio of 70–30% (GL(30)). On the other hand, Zn 2*p*, O 1*s*, and N 1*s* spectra were fitted by only symmetric Gaussian−Lorentzian convoluted functions with a ratio of 70–30% (GL(30)).

The measurements of the O *K*-edge, N *K*-edge, and C *K*-edge spectra were carried out in the total electron yield mode, yielding a probing depth of ca. 10–15 nm [30], and at a “magic” angle of 54.7°, to provide a comparable contribution of the π- and σ-related states. The as-recorded XAS spectra were conventionally normalized and smoothed using the algorithm described in [31].

### 2.5. Gas-Sensing Studies

The gas-sensing performance of the fabricated prototype chips composed of AmG and Am-ZnO layers was studied by collecting their multisensor chemiresistive responses while exposed to a set of test analytes comprising ammonia, ethanol, and water vapors mixed with dry air. These particular analytes were chosen because, on the one hand, they are among the most representative test gases to compare the performance of the developed multisensor chips with the published data and, on the other hand, their detection is highly important in industry and ecological monitoring. In the case of the dry air, the humidity level was maintained below 200 ppm, with other gaseous contaminants ensured with a dry air generator (PG14L, Peak Scientific, Glasgow, UK) at levels below 0.1 ppm [27].

All of the analytes were generated by bubbling corresponding solutions of analytical purity with air flow to obtain saturated vapors, which were subsequently mixed with the pure air. Managing the ratios between the saturated vapors of the analytes and the air, the analyte probes were brought to various concentrations of 500–10,000 ppm to be delivered to the chips under study. The gas-mixing setup was assembled from a compressor (Peak Scientific, Glasgow, UK) with a set of pipelines equipped with two-way and three-way remote valves and driven by a personal computer (PC), as displayed in the Appendix A. In the case of the measurements with humid air, the humidity was maintained at the level of 25 rel. %. by the addition of air flowing through the bubbler containing distilled water.

Prior to the measurements, the chips under study were mounted into a 50-pin ceramic holder (Siegert, Cadolzburg Germany) with an Erni SMC 1.27 mm connector. Mounting was carried out via ultrasonic welding of Al wires (38 µm diameter) by a bonder (74767, West Bond, Anaheim, CA, USA). The chips in the holder were then housed in a chamber composed of two stainless steel halves with heat-resistant sealing rings and two tube entries for gas flow input and output. Screwed to each other, the chamber halves fixed the chip in the holder between them, forming an isolated gas probe volume of ca. 0.2 cm^3^, while the multi-pin connector stayed out. For each concentration of the analyte, namely, 500 ppm, 1000 ppm, 4000 ppm, and 10,000 ppm, the chips were exposed for 30 min followed by purging the chamber with a background air for 60 and 90 min for the tests at room temperature and upon heating, respectively. This procedure was repeated for each analyte. To provide independent control of sensor response and to ensure the analyte’s delivery, the gas flow out of the chamber was forwarded through a line equipped with a capacitive humidity sensor (ASAIR AM2302, Aosong Electronics Co., Guangzhou, China) and a semiconductor sensor (MQ-3, Hanwei Electronics Co., Zhengzhou, China). The signals from these sensors were also read and stored on a PC using the Arduino Nano based on an ATmega 328p chip (Atmel, San Jose, CA, USA).

To read out the multisensor chemiresistive responses provided by each pair of electrodes with a confined layer of AmG or Am-ZnO, the home-made measuring unit was designed by taking a data acquisition platform (NI-DAQ, National Instruments, Austin, TX, USA), proportional–integral–derivative (PID) controller, and a multimeter (Keithley 2000, Solon, OH, USA), all connected to a PC via RS-232/USB interfaces to store/visualize and to subsequently process the data. The resistance values were acquired with the help of a high-precision multimeter by sequential gauging of the electrode pairs via multiplexing relays driven by an Arduino circuit interfaced to a PC via USB. Prior to the *R*(*t*) measurements, the *I*-*V* curves were recorded at a [−1;+1] V range to check the ohmic behavior of the contacts.

A set of prototype chips was prepared and studied at different stages of the current investigation. Here, the data for two exemplary chips are reported, which were consistently and systematically measured under the described conditions. The gas-sensing studies for both chips composed of AmG and Am-ZnO layers at the first stage were carried out at RT to be conditioned at 25 ± 5 °C. For the Am-ZnO on-chip multisensor array, additional measurements were carried out at elevated temperatures in the range from 60 °C up to 310 °C, with a step of 50 C at the second stage. The heating was provided by a pair of on-chip thermoresistors controlled by on-chip meander-shaped heaters to be connected to the PID controller. The thermoresistors were calibrated by an IR camera (R500EX-P D AVIO/NEC, Nippon Avionics, Kanagawa, Japan) with 21 μm spot resolution.

The chemiresistive response of the sensor elements was calculated as follows:(1)$$S=\frac{R-R_{air}}{R_{air}}\cdot 100\%$$
where *R* and *R_air_* denote the resistance values recorded upon the analyte vapor’s appearance and the background air right before each gas pulse as a reference, respectively [32]. At the same time, the recorded *R*(*t*) transients are displayed in the form of *R*/*R*_0_, where *R*_0_ is the resistance of the sensing element prior to the whole sequence of pulses for each analyte. This allows one to explicitly evaluate and compare both the chemiresistive responses and the total resistance drift upon exposure to different analytes.

Upon processing, the data from the 21 sensing elements, with similar thicknesses of the AmG/Am-ZnO gas-sensing layer on both chips, were employed to provide a representative comparison of the chemiresistive properties of these materials. The multisensor signals from the chips under study were processed by the pattern recognition technique via linear discriminant analysis (LDA), employing home-made software with the purpose of distinguishing data clusters related to various analytes [33]. Within the frame of this method, the raw multisensor signals upon exposure to analytes are denoted as classes and transferred to a reduced artificial space of features (LDA space) by means of estimating a maximum ratio between inter-class and intra-class variations. For the LDA analysis, the resistance values of all sensing elements under quasi-stationary conditions (*R_i_*), recorded upon exposure to 4000 ppm of each analyte in the dry and humid air, were taken for the analysis. Given the generated LDA spaces, the gravity center for each class was determined and further distanced by Mahalanobis metrics—that is, the Euclidian distance between gravity centers of the class-related clusters [34]. The distinctiveness of the multisensor data related to each of the five analytes plus pure air was estimated as an averaged distance between the coordinates system center and gravity centers of the class-related clusters. For illustration purposes, 3-dimensional cross-sections of the total LDA space were employed.

## 3. Results

### 3.1. AmG and Am-ZnO Morphology

Figure 1 exhibits the results of the electron microscopy studies of the AmG prior to and after the deposition of the ZnO NPs. The initial AmG appeared as highly corrugated flakes with a large number of folds up to several micrometers in height (Figure 1a). This morphology, considerably distinct from the lamellar GO flakes [35], is an inherent feature of the aminated graphene, being observed regardless of the deposition method or the solvent used. It stems from, on the one hand, the loss of basal-plane oxygen groups, forcing the graphene layer to be flattened due to their electrostatic repulsion [36], and on the other hand, distortion of the graphene lattice by amine groups, which tend to be clustered in rows and promote the formation of folds [23]. The latter factor is reflected by the acquired TEM images and ED patterns, examples of which are displayed in Figure 1b. A set of distinguishable wrinkles that were not initially present in GO (Appendix A) but arose upon the amination were observed to protrude through the AmG layer. This was accompanied by the corresponding ED pattern becoming blurred, albeit still presenting as a single set of six distinct (10) and (11) diffraction maxima. This latter fact indicates the predominantly monolayer nature of the synthesized AmG. The observed blurring of the diffraction maxima compared to the ones for the initial GO (Appendix A) arose from the alterations in the angle between the electron beam and the AmG surface [37,38], signifying the nanoscale wrinkling of the material.

Similar to AmG, Am-ZnO preserves the tendency to form corrugated structures, with arguably even higher density of folds, as shown in Figure 1c,d. At the same time, the arrays of ZnO NPs covering the AmG layers were observed amidst the folds, as most clearly displayed in Figure 1e. Notably, ZnO NPs were homogeneously distributed on the surface of AmG, with the absence of either indicatable areas ZnO aggregates or void areas without NPs. This claim is further supported by the collected TEM images, a pair of which is displayed in Figure 1f,g. According to the acquired images, ZnO NPs with an average diameter of ca. 8 nm (Appendix A) were uniformly arranged throughout the AmG layer, comprising groups of 3–4 NPs at maximum, which were still isolated from each other without the formation of the percolation network. The mean distance between the NPs and their assemblies was ca. 10 nm, whereas the surface coverage percentage was estimated to be 29.5%. This drastically differs from the rGO-ZnO composites reported previously [6,39,40] and those fabricated within the scope of this work via the same procedure (Appendix A). Large clusters of ZnO were observed to be separated by areas of free graphene layers with few-to-no individual ZnO groups. The same morphology was observed in the case of taking lower ZnO contents in the suspension in an attempt to reduce the degree of aggregation.

Thus, employing AmG advances the formation of Gr-ZnO nanocomposites with arrays of evenly distributed individual ZnO NPs, due to the advantageous role of amine groups, which coordinate and immobilize NPs through non-covalent bonding to be further manifested by the spectroscopic studies.

### 3.2. Core-Level Studies of the AmG and Am-ZnO

High-resolution TEM imaging (Figure 1h,i), along with the ED studies (Figure 1j), revealed that the ZnO NPs had a hexagonal structure (wurtzite-type, space group P6_3mmc) faceted predominantly by (0001), {−1100}, and {−1101} planes [41,42]. To further assess the chemistry of Am-ZnO and of the initial AmG, we employed core-level techniques, namely, XPS and XAS. Figure 2a exhibits the survey spectra of the materials under study, which indicate the absence of any contaminating impurities in both graphene derivatives. This fact is reflected by the presence of exclusively C 1*s*, N 1*s*, and O 1*s* core-level signals centered at binding energies (*BE*) of 284.7 eV, 400.1 eV, and 532.5 eV, respectively, for the AmG, which are accompanied by a set of Zn-related lines, including a Zn 2*p* doublet at *BE*s of 1045.3 eV and 1022.1 eV, along with Zn LMM, Zn 3*s*, Zn 3*p*, and Zn 3d lines centered at 212.4 eV, 143.8 eV, 91.8 eV, and 13.2 eV, respectively [43,44]. Given the collected survey spectra, the elemental composition of the AmG and Am-ZnO samples was evaluated. Figure 2b displays the corresponding plot bars. The performed amination of GO was found to result in the introduction of up to 7.74 at.% nitrogen, corresponding to the higher reported values for the aminated graphenes [45,46,47,48]. In turn, introducing ZnO NPs led to the appearance of Zn at a concentration of 5.61 at.%, with a corresponding redistribution in the other elements’ contents. Notably, an increase in oxygen concentration from 5.38 at.% to 9.67 at.% was indicated, along with a reduction in the nitrogen and carbon contents down to 5.27 at.% and 79.45 at.%, respectively. The almost perfect matching between the concentrations of Zn and O suggests negligible content of metallic Zn, as well as oxygen vacancies (OVs) in the ZnO nanoparticles [49].

This assertion was well supported by the subsequent examination of the Zn 2*p* and O 1*s* lines after deconvolution, as displayed in Figure 2c,d. Three doublets can be discerned in the Zn 2*p* line. The first one, dominant in the spectrum and centered at *BE*s of 1022.2 eV (Zn 2*p*_3/2_) and 1045.1 eV (Zn 2*p*_1/2_), with spin–orbit splitting (Δs) of 23.0 eV, is well known to correspond to Zn in the divalent oxidation state (Zn^2+^ ions) in Zn-O bonds of the wurtzite structure or Zn-OH bonds of the Zn(OH)_2_ surface layer [50,51,52]. In turn, a pair of other doublets with *BE*s of 1020.3 eV and 1042.8 eV, and of 1024.3 eV and 1046.8 eV, respectively, with Δs of ca. 22.5 eV, matured from the metallic zinc (Zn^0+^) in the former case and OVs in the ZnO NPs in the latter one [53,54,55]. However, the relative contents of Zn in these forms were considerably low, estimated to be 11.6% and 5.6%, respectively, indicating that the nanoparticles were mainly constituted by ZnO and Zn(OH)_2_, contributing up to 82.8%.

The presence of both ZnO and Zn(OH)_2_ was signified by the deconvolution of the O 1*s* spectrum, as displayed in Figure 2d. Three peaks were discerned, namely, Zn-O_w_ centered at a *BE* of 529.6 eV, Zn-OH with a *BE* of 531.2 eV, and O-C/O-N at 532.5 eV. The first spectral components corresponded to O^2−^ ions participating in the Zn-O bonding of the wurtzite structure of ZnO NPs [41,50,56]. The nature of the second peak is more disputable. Commonly, it is attributed to oxygen vacancies in ZnO nanoparticles [16,57], which, if located on the surface of the ZnO nanoparticles, form dangling bonds. However, Frankcombe et al. found this to be a misinterpretation, instead attributing the spectral component at these BEs to the OH-related species [58]. Given that the particular relation between the Zn-OH and Zn_(OW)_ contributions could be derived neither from Zn 2*p* nor from other XPS or XAS data, we attributed this peak to both possible states in the ZnO NPs. The last spectral component, lying around 532.7 eV, was attributed to O-C bonds in the Am-ZnO layers as well as adsorbed oxygen in the ZnO NPs. The derived relative contributions of the distinguished spectral components were 60.7%, 29.5%, and 9.8% for the Zn-O_w_, Zn-OH/Zn_(OV)_, and O-C/O_ad_, respectively. Such a ratio indicates a prominent contribution of the −OH and Zn_(OV)_ species on the surface of the NPs.

The acquired O *K*-edge XAS and XPS data on the chemistry of ZnO NPs are well supported by the spectra exhibited in Figure 2e. Despite the almost complete absence of oxygen groups in the AmG, a low-intensity but informative spectrum was collected, highlighting the presence of the retained carboxyl (COOH) and ketone (C=O) groups by the presence of π* and σ* resonances related to C 1s–π* or –σ* transitions in these moieties. Particularly, the XAS spectrum yielded π* (COOH) resonance at a photon energy (PE) of 531.3 eV, accompanied by σ* (O-H) and σ* (COOH) resonances around PEs of 534.7 eV and 544.8 eV, respectively, owing to retained carboxyls. In turn, a pair of π* (C=O) and σ* (C-O) resonances centered at PEs of 532.5 eV and 540.1 eV, respectively, revealed the incomplete elimination of ketones upon the amination [59,60].

The introduction of ZnO NPs enhanced the intensity of the O K-edge XAS spectrum, along with its substantial transformation expressed in the appearance of a set of new resonances. Notably, a dominant feature at a PE of 540.7 eV, accompanied by a shoulder at a PE of 543.9 eV, was observed, which corresponds to electron excitation from the O 1 s orbital to the hybridized O 2pσ-Zn 4sp orbitals and O 2pπ-Zn 4d orbitals, respectively [61,62,63]. These resonances are typical for all ZnO NPs, irrespective of their type. However, the appearance of a high-intensity O 2pσ-(Zn 4s)_w_ resonance around a PE equal to 534.3 eV is a characteristic feature of the wurtzite-type ZnO NPs, as was pointed out by Cho et al. [62]. Furthermore, distinguishable O 2pπ-(Zn 4sp)_OH_ resonance centered at a PE of 540.7 eV was indicated, being related to the transitions from O 1 s orbitals to hybridized O 2pπ-Zn 4sp orbitals of the -OH groups. The appearance of this resonance verifies the formation of the hydroxide surface layer on the top of ZnO nanoparticles. Thus, XPS and XAS data support the TEM results on ZnO nanoparticles as wurtzite-type, but they also specify the presence of a hydroxide surface layer with valuable contents of -OH species.

In addition to specifying the chemistry of ZnO NPs, the collected XPS and XAS data also hint at details regarding bonding between them and the AmG layer through a non-covalent interaction with the amine groups. Still, the exact structure of amine molecules over the ZnO surface has not been studied systematically. Early reports indicate that the amine group bonds with Zn atoms on the ZnO surface [64,65]. The high-resolution N 1*s* spectra of AmG and Am-ZnO displayed in Figure 2f indicate no significant changes in the forms of nitrogen upon the deposition of ZnO. Particularly, no signs of the Zn-N peaks found commonly around 395–396 eV can be observed [66]. This implies an absence of covalent bonding between ZnO NPs and amine groups through the following possible reaction:Gr-NH_2_ + Zn-OH → Gr-NH-Zn + H_2_O(2)

We attributed this to the fact that such a condensation reaction is unfavorable in aqueous media or in humid air at room temperature. In turn, a redistribution in the integral intensities of the spectral components centered at *BEs* of 400.1 eV and 401.5 eV was indicated: the contribution from the former dropped from 71.83% to 52.27%, while for the latter it rose from 9.62% to 30.16%. The first of these spectral features is known to stem from the presence of primary and secondary amines [22,23], whereas the origin of the second one is more ambiguous. Commonly, it is attributed to the presence of graphitic nitrogen embedded into the graphene network [67]. However, no sample annealing at high temperatures or treatment with harsh chemicals was carried out, which could have caused the amines to convert into graphitic nitrogen [45]. Furthermore, no signs of π* resonance, corresponding to graphitic nitrogen and generally found at *PEs* of 400.5–400.8 eV [45,67], can be observed in the acquired N *K*-edge XAS spectra displayed in Figure 3a. Only prominent π* resonance of the amines centered at *PEs* of 401.2 eV–401.7 eV is present, accompanied by π* resonance of pyridines at a *PE* of 398.3 eV [22,68].

At the same time, it was demonstrated that the presence of the protonated amines, NH_3_^+^, also gives rise to the peak in the N 1*s* spectra, with BEs of 401.3–401.6 eV [69,70]. This difference in the peak position compared to that of amine is due to the reduction in the electron density on the nitrogen atom induced by protonation and, consequently, the higher *BE* for the corresponding peak. Thus, the observed enhancement of the peak at a *BE* of 401.5 eV, together with the simultaneous reduction in the amine peak at 400.1 eV, can be asserted to indicate a protonation of a considerable part of the amines in the AmG. This is further justified by a more precise look at the N *K*-edge XAS spectra. Similar to the XPS case, the protonation of amines should induce the shift of the corresponding π* resonance towards higher *PE*s [70]. The corresponding effect was indicated for the π* resonance of amines upon moving from AmG to Am-ZnO, which shifted from 401.2 eV to 401.7 eV, while the position of the π* resonance of pyridines remained unaltered. The signified protonation of amines was finally confirmed by the acquired C 1*s* XPS spectra, as shown in Figure 3b. For both materials, the spectra were dominated by the C=C peak of the pristine graphene network at a *BE* of 284.7 eV, with a negligible contribution of oxygen-related spectral features at *BE*s of 286.8 eV (not distinguished), 288.1 eV, and 289.0 eV, derived from the presence of basal-plane hydroxyls and epoxides, ketones, and carboxyls, respectively [45,59,71]. Furthermore, no valuable contribution of the C-V peak centered at 283.8 eV, appearing from the dangling bonds of carbon atoms [71], was revealed. This fact emphasizes the absence of dangling bonds in the graphene network at valuable contents that could affect the properties of the AmG and Am-ZnO layers, particularly their gas-sensing performance.

Apart from the aforementioned spectral features, a prominent C-NH_2_ peak with a *BE* of 286.1 eV can be discerned in the AmG spectrum, which shifts to a lower *BE* of 285.7 eV in the case of Am-ZnO. Such alterations hint once again at the reduction in the electron density on the nitrogen atom as a sequence of the amine protonation. Although the peaks of basal-plane hydroxyls and epoxides, as well as edge-located hydroxyls, are known to contribute to the discussed range of *BE*s, possibly affecting the position of the C-NH_2_/C-NH_3_^+^ peak, the effect of these features is negligible because no signs of their presence can be discerned in either the C 1s XPS spectra or the C *K*-edge XAS spectra (Figure 3c). No π* resonance can be observed for edge-located hydroxyls around a *PE* of 286.5 eV, nor can epoxide- and hydroxyl-related resonances of the basal-plane epoxides and hydroxyls at *PE*s of 287.3 eV and 289.6 eV, respectively [59,72]. In turn, prominent π* (C=C) resonance at a PE of 285.1 eV and σ-exciton at a PE of 291.65 eV are clearly present, representing characteristic signs of maximal elimination of the oxygen groups and a high degree of graphene network recuperation, even in the presence of up to 5–7 at.% concentrations of amines.

These findings allow us to derive the bonding mechanism between AmG and ZnO NPs. Wurtzite-type ZnO NPs are known to exhibit a positive net charge [73], preventing them, at first glance, from effective interaction with the positively charged AmG surface. However, aminated graphene behaves as a base in aqueous media, advancing its pH value to 9–10. This causes the hydroxide surface of ZnO nanoparticles to donate protons and form negatively charged surfaces (≡M–O^−^) [74]. Accordingly, a double-charged system appears, consisting of positively charged protonated amine groups on the AmG surface on one side and negatively charged ZnO NPs on the other. As a result, strong electrostatic bonding between AmG and ZnO appears, leading to stable immobilization of the latter due to the high abundance of the participating amine groups.

The efficiency of such bonding was determined by examining the morphology of the Am-ZnO layer after its high-temperature annealing at 315 °C for 3 h. Figure 3d–g exhibit exemplary SEM and TEM images collected for the Am-ZnO layers following the heat treatment. No substantial aggregation can be observed in the acquired images. The ZnO NPs continued to be arranged individually or in small groups of 3–5 particles distributed evenly throughout the AmG layer. Importantly, no percolating chains appeared with NPs or their groups, causing them to remain isolated from each other at an inter-particle distance of 5–10 nm. This drastically contrasts with the aggregation of ZnO NPs on the surface of rGO (Appendix A), where large clusters of 50–100 nm in size were observed, accompanied by the formation of a percolation network. A set of rGO-ZnO samples was examined to confirm the invariable aggregation of ZnO NPs and the formation of their chains along the graphene layer.

As a net result, graphene amination was found not only to benefit the uniform and almost-ordered distribution of the ZnO NPs over the graphene layer, but also to provide an efficient immobilization of the NPs, stable up to 300 °C heating, thanks to the formation of a network of ionic bonds governed by the abundance of the protonated amine groups.

### 3.3. Gas-Sensing Performance of the on-Chip Multisensor Arrays Comprised of the AmG and Am-ZnO Layers

Given the information on the morphology and chemistry of AmG- and Am-ZnO, both materials were tested as a gas-sensing layer for the multisensor e-nose chips applied over the Si substrate equipped with multiple co-planar electrodes (see Materials and Methods). Figure 4a exhibits an optical photo of the developed chip composed of the AmG layer, exhibiting the same morphology as the one based on Am-ZnO. Figure 4b,c display the morphology of the AmG and Am-ZnO layers, respectively, in their thinnest parts. As one can see, the deposited films in both cases are presented as arrays of overlapping Am/Am-ZnO flakes, 1–10 μm in size, forming a conductive layer to cover the substrate and the measuring electrodes. Notably, the wrinkling of the flakes discussed in previous sections can be clearly seen, yielding a higher specific surface area of the gas-sensing layer. To verify the ohmic contact between the graphene-based layers and the measuring electrodes, the current–voltage characteristic (*I*-*V*) was recorded in dry air (Figure 4d,e). For the AmG layer, a linear *I*-*V* curve was observed, whereas slight non-linearity with a mismatch between the forward and backward scans was observed in the Am-ZnO case. We assumed that this was related to the formation of the Schottky barrier at the hybrid interface between the ZnO NPs and the AmG layer due to the work function differences (3.37 eV and 4.3 eV, respectively [14,23]), which further significantly separated the charge carriers and promoted the spatial charge transfer via the built-in electric field [15]. This fact is also reflected by a significant (more than one magnitude) enhancement of the layer’s average resistance upon moving from AmG to Am-ZnO, from ca. 370 Ω to ca. 8000 Ω, as observed by comparing the *I*-*V* characteristics of the materials under study. Thus, the acquired *I*-*V* curves imply the absence of significant potential barriers at the interface between the gas-sensing layer and the measuring electrodes, while also indicating the possible formation of an Am-ZnO p-n heterojunction interface, enhancing sensitivity towards gas molecules.

To examine and compare the gas-sensing performance of the chips under study, they were primarily exposed to several analytes, particularly ammonia (NH_3_), ethanol (C_2_H_5_OH), and water (H_2_O) vapors, in a concentration range of 500–10,000 ppm, in mixtures with dry air at RT. The chips were exposed to the analytes for 30 min, with intermediate purging by dry air for 60 min to reach the proper saturation in the sensor responses and to collect samples for the array’s vector signals. Figure 4f–h display the *R*(*t*) transients of the exemplary sensors of the multisensor arrays based on AmG and Am-ZnO layers with the same thickness of ca. 160 nm.

For both materials, a distinct increase in the exemplary sensors’ resistance upon exposure to all of the named analytes was indicated, with a reversible drop almost to the initial *R*_0_ under the subsequent purging with air according to a typical gas sensor’s operation. Even after being exposed to analytes at high concentrations of 10,000 ppm, graphene-based chips recover almost entirely after each gas exposure pulse, without the need for UV irradiation or annealing at high temperatures of 200–350 °C, which are commonly employed to regenerate gas sensors during their operation [7,8,75]. The total drift of the *R*_0_ upon the whole sequence of the pulses was below 0.79% in the case of the AmG layer and 2.46% in the case of the Am-ZnO layer, reaching the maximum in NH_3_ vapors and almost zero in H_2_O vapors for both materials. Water vapors even reduced the *R*_0_ in the case of Am-ZnO, probably due to a more prominent reduction in the electron-withdrawing effect of the ZnO NPs as a result of the desorption of H_2_O molecules upon purging with air [76]. The more pronounced shift in the *R*_0_ in the case of Am-ZnO also suggests the aforementioned possibility of oxygen vacancies at the surface of the ZnO NPs, leading to the trapping of the gas molecules. Notably, the change in the *R*_0_ during the measurements was slow and monotonic, so it could be readily compensated by routine signal processing with an analog-to-digital converter, to be integrated into the package board of the multisensor chip in the commercial implementation of this system.

The chemiresistive response of the AmG and Am-ZnO layers is reproducible; magnitudes (S) are found to depend on the concentration of the analytes according to the Freundlich isotherm described by the *S*~*C*^α^ function, as displayed in Figure 5a. At the same time, the substantially pronounced chemiresistive response in the case of Am-ZnO compared to AmG was clearly signified for all of the analytes. The magnitudes of resistance changes towards 500 ppm of NH_3_, C_2_H_5_OH, and H_2_O were 0.11%, 0.04%, and 0.08%, respectively, for AmG, whereas these values increased by more than an order of magnitude to 1.25%, 1.37%, and 1.72%, respectively, in the case of Am-ZnO. These results indicate that arrays of ZnO NPs substantially enhance the gas-sensing performance of the aminated graphene. We assert that this effect originates from the formation of a net of p-n heterojunctions at the interfaces of the AmG and ZnO NPs, which are known to facilitate the gas sensitivity of carbon-based materials [77,78,79]. This is supported by our observation of such an enhancement of the chemiresistive response at RT, while metal oxides, even at nanoscale, require heating up to T = 200–300 °C to operate. Furthermore, no considerable changes in the response and recovery times were observed, being equal to ca. 2 min and ca. 7 min, respectively, toward C_2_H_5_OH and H_2_O, for both AmG and Am-ZnO. The observed similarity in the response and recovery times implies that the adsorption processes proceed mainly at the surface of AmG, not the ZnO nanoparticles, possibly at the p-n heterojunctions. Given this, the even distribution of arrays of individual NPs, yielding a higher number of p-n heterojunctions, along with their strong electrostatic interaction with the graphene layer through amine groups, which enhances the effect, can be said to play a key role in promoting the chemiresistive response of the material [77].

Moving to 10,000 ppm, the response values increased to 0.4% (NH_3_), 0.29% (C_2_H_5_OH), and 0.46% (H_2_O) for AmG and to 4.5%, 5.4%, and 7.36% for Am-ZnO layers, with the corresponding power coefficient, α, estimated to be 0.42, 0.61, and 0.57 for the former graphene derivative and 0.45, 0.43, and 0.49 for the latter one, respectively. Apparently, the raw signals of several percent and even tens of percent do not fit almost any of the practical applications [7,8]. However, the observed exceptional robustness towards high concentrations of analytes is an advantage for employing such materials in gas leak detectors, which are in high demand in the case of ammonia. Furthermore, regarding the use of signal-to-noise ratio instead of raw chemiresistive signals, as a conventional way to process the acquired electrical characteristics, the valuable sensitivity of the multisensor chips under study was demonstrated. Figure 5b exhibits plotted *SNR* values versus the concentrations of the studied analytes. Similar to the S(C) curves, the *SNR*(C) trends follow the power law SNR~C^α^, with α values of 0.43, 0.41, and 0.51 for NH_3_, C_2_H_5_OH, and H_2_O, respectively, while the SNR values for the corresponding analytes at 500 ppm were estimated to be 19.2%, 25.0%, and 50.4%, respectively. Extrapolating the acquired *SNR*(C) towards the cap of *SNR* = 3, corresponding to the minimum practical level of signal detection, the limit of detection (LOD) was determined to be 5.1 ppm, 3.6 ppm, and 1.3 ppm for NH_3_, C_2_H_5_OH, and H_2_O, respectively. These values are among the most commonly reported for state-of-the-art graphene-based gas-sensing devices (Appendix A) and already fit almost all practical demands for the detection of ammonia and alcohols [7,8,14,15,16,18,27,79,80,81].

To further manifest the operation of the fabricated multisensor chips under various conditions, the chemiresistive response of the on-chip sensor arrays composed of AmG and Am-ZnO layers toward NH_3_ and C_2_H_5_OH mixed with humid air was recorded and examined. Figure 5c,d display the acquired *R*(*t*) transients, which indicate that the humid air substantially reduced the chemiresistive response of Am-ZnO and enhanced the noise level. This was assumed to originate from water molecules readily adsorbing on ZnO NPs and at p-n heterojunctions, enhancing the base resistance of the Am-ZnO layer and hindering the adsorption of the analyte molecules (NH_3_ or C_2_H_5_OH). This resulted in a lower chemiresistive response towards these analytes, along with the increased noise level due to constant adsorption–desorption of water and analyte molecules.

Nevertheless, chemiresistive signals of both materials, AmG and Am-ZnO, were still observed along the whole concentration range, although they dropped by almost twofold in Am-ZnO, with the magnitude of the resistance change towards 500 ppm of NH_3_ and C_2_H_5_OH being reduced to 0.61% and 0.69%, respectively. Accordingly, the *SNR* values decreased to 13.6% and 10.2%, respectively, while the LOD increased to 63.1 ppm and 83.4 ppm, respectively. Despite not being as low as for dry air, these values still meet the requirements for many practical applications; for instance, the limit of exposure to ethanol vapors noted by the US Occupational Safety and Health Administration (OSHA) is 1000 ppm [82]. Interestingly, unlike the case of Am-ZnO, almost no changes were indicated in the gas-sensing performance of the AmG layer in humid air, with the signal toward 500 ppm of NH_3_ and C_2_H_5_OH remaining around 0.4% and 0.29%, respectively. The difference appeared only at high concentrations of 4000–10,000 ppm, with the reduction in the signal by ca. 1.5-fold, which is a known phenomenon for graphene-based gas sensors operating in humid air [27,83]. At the same time, no enhancement was indicated in the chemiresistive response toward NH_3_ in humid air, which was demonstrated previously for other graphene derivatives (carbonylated and carboxylated graphenes) [27,84], highlighting a unique feature of these functionalized graphenes.

In addition to the sensitivity, selectivity is another key parameter of the gas-sensing units, which is often harder to achieve. Analytes of different kinds, such as alcohols and ammonia, commonly induce a chemiresistive response of the same magnitude due to similar mechanisms involved in their interaction with the gas-sensing material [10,85,86]. This was also seen for the chips under study from the plotted chemiresistive responses against the studied analytes at an exemplary concentration of 4000 ppm in dry air, as displayed in Figure 5e. All of the analytes provided forward chemiresistive responses of the same magnitude, making it impossible to distinguish them by a single sensor element. However, the test gases and VOCs could be reliably discerned by taking advantage of the recording vector signals generated by multiple sensor elements in the on-chip multisensor array. Figure 5f exhibits characteristic radar patterns of the raw vector signal, S = {S_i_}, generated by all sensing elements and corresponding to the distribution of the chemiresistive response along the on-chip sensor array composed of AmG and Am-ZnO, as recorded upon exposure to the vapors of interest. Data for 21 sensing elements located in the central part of the chip showed a comparable thickness of the deposited AmG and Am-ZnO layers, as well as less noisy signals. The displayed patterns are distinctive to the naked eye, not only in the case of comparing AmG and Am-ZnO, but also in the case of various analytes while considering a single gas-sensing material. Notably, the vector signal in the case of AmG differed more from analyte to analyte, which probably stems from more alterations in the morphology and pore structure across the AmG layer compared to Am-ZnO, where NPs are uniformly arranged on the AmG platelets.

Given such distinct vector signals for the AmG and Am-ZnO layers, reliable identification of the analytes can be expected via further processing of the acquired data with a pattern recognition technique such as LDA. Within the framework of this approach, each multisensor vector signal recorded from the chip while exposed to a certain analyte is then transferred to the artificial space and processed to find the maximum inter-class variations related to intra-class fluctuations [27,84,86]. Figure 6 exhibits the results of LDA processing with the first three LDA components being present. The points correspond to processed vector data for a single measurement, which can be seen to be grouped into clusters, filling analyte-related ellipses with a radius defined by a 0.9 confidence level under the suggestion of Gaussian distribution of data within the class. For both AmG- and Am-ZnO-based chips, the analyte-related clusters are well distanced from each other, as well as from the signal of the background air. The averaged distances between the central coordinates the and analyte-related clusters were established to be 124.8 arb. units for the AmG-based chip and 219.6 arb. for the Am-ZnO one. These results indicate that the developed chips composed of both AmG and Am-ZnO are applicable to recognize and selectively detect all of the studied analytes in both dry and humid air.

To finally assess the role of ZnO NPs in enhancing the chemiresistive response and further examine the performance of the Am-ZnO chip in various practical conditions, its gas-sensing properties for detecting analytes at elevated temperatures in the 303–583 K range were probed in the second stage of our experiments. Figure 7a–c exhibit the acquired *R*(*t*) transients upon exposing the chip to two pulses of NH_3_, C_2_H_5_OH, and H_2_O, at 5000 ppm concentrations. Only a slight enhancement in the signal level towards NH_3_ was observed upon the increase in temperature (Figure 7d), although a considerable reduction in the noise level was indicated. However, more interesting behavior was noted for C_2_H_5_OH and H_2_O. The chemiresistive response dropped rapidly upon heating up to 110 °C, reaching minimum values of 1.65% and 1.4% for C_2_H_5_OH and H_2_O, respectively. This was followed by a progressive backward growth up to 310 °C, without yet achieving the initial signal values. Given these results, with no substantial enhancement of the chemiresistive response upon heating, typical for metal oxide gas sensors [33,87], the key role of *p*-*n* heterojunctions was further signified. The origin of the discovered reduction in the chemiresistive response during the first steps of annealing remains unclear, probably being related to the rearrangement of the hydroxide layer of ZnO NPs and their interaction with the AmG layer.

This was indirectly evidenced by the subsequent examination of the chip’s gas-sensing properties after its annealing at the aforementioned temperatures. Figure 7e–g show the *R*(*t*) transients recorded upon exposing the Am-ZnO chip to all analytes in dry air after annealing. Firstly, the twofold reduction in the *R*_0_ value was found to be accompanied by a noticeable diminution in the *R*_0_ drift, which disappeared almost completely for the C_2_H_5_OH and H_2_O measurements. Furthermore, enhancement in the sensitivity to both of these analytes was noted, with an increase in the chemiresistive response towards exposure to 500 ppm from 1.3% and 1.7% up to 2.3% and 2.1% for C_2_H_5_OH and H_2_O, respectively. In contrast, almost no changes in the sensitivity to NH_3_ were indicated. We assert that these changes in the gas-sensing performance towards C_2_H_5_OH and H_2_O originated from the complete elimination of the water molecules retained to be adsorbed after the chip’s fabrication and following the first stage of gas-sensing measurements. These molecules lock and reduce the number of feasible adsorption sites for analytes. This is also supported by the indicated decrease in the *R*_0_ value, which should stem from the same cause as well as the aforementioned assumed rearrangement of the hydroxide layer on the ZnO NPs with the reduction in the electron-withdrawing effect induced by them. At the same time, the absence of any lessening of the gas-sensing response, which even increased after the high-temperature treatment, proves the high stability of the Am-ZnO structure achieved due to the immobilization of ZnO NPs of amine groups, once again highlighting the advantageous properties of the synthesized graphene-based nanocomposite.

## 4. Conclusions

In summary, we established the beneficial role of graphene amination for the fabrication of Gr-MO composites and, in turn, an advantageous effect of the introduced ZnO NPs on the gas-sensing properties of the aminated graphene. A uniform distribution of ZnO NPs arranged into arrays of individual and few-particle clusters of NPs, which are isolated from each other and do not form a percolation network, can be achieved by employing the aminated graphene. Conversely, using a conventional rGO layer without amine groups was shown to result in ZnO NPs being aggregated on the surface of the graphene layer, forming large clusters and percolating chains.

Complementing the electron microscopy techniques with XPS and XAS studies, it was further revealed that the beneficial effect of amines originates from their protonation upon interacting with the hydroxylated surface of ZnO NPs. This results in their efficient immobilization through a network of ionic bonds. The exceptional stability of the designed nanocomposite was justified by the absence of any considerable aggregation of ZnO NPs even after high-temperature annealing at temperatures of up to 315 °C, which corresponds to the highest operating temperature for such composites in practical applications. Moreover, fingerprints of the ZnO and protonated amines in the XPS and XAS spectra were refined. This extends the applicability of these methods to examining the chemistry of ZnO-based structures and amine-containing compounds.

On-chip multisensor arrays composed of the AmG and Am-ZnO layers were successfully fabricated and examined, displaying the immobilization of ZnO NPs to facilitate the gas-sensing properties of the aminated graphene. Notably, the chemiresistive response in the Am-ZnO layer was shown to be enhanced by one order of magnitude when compared to values of the AmG layer, reaching the LOD of several ppm for NH_3_, C_2_H_5_OH, and H_2_O in dry air under RT conditions. Combined with the non-linear character of the *I*-*V* curves and the absence of considerable alterations in the chemiresistive response upon heating the Am-ZnO up to 310 °C, these results indicate that the gas-sensing properties originate from the formation of arrays of p-n heterojunctions between the aminated graphene and ZnO NPs. However, further theoretical studies should be performed to specify their band structure modulation upon the adsorption of gas molecules.

At the same time, even higher chemiresistive signals were indicated after high-temperature treatment. On the one hand, this confirms the absence of the ZnO nanoparticles’ aggregation, and on the other, it implies a complete elimination of the adsorbed water, which hinders the sensing properties toward alcohol and water vapors. Finally, a distinct multisensor response toward all of the analytes, in both dry and humid air, was demonstrated, allowing us to selectively identify the studied analytes by employing combinatorial processing of the vector signal via a pattern recognition technique like LDA. We consider these materials to extend the available options to develop power-reduced units for environmental monitoring and Internet-of-Things applications.

## Figures and Tables

**Figure 1 nanomaterials-14-00735-f001:**
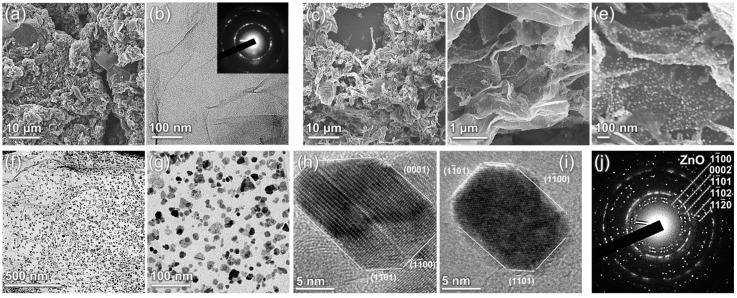
Electron microscopy studies of the AmG and Am-ZnO: (**a**) SEM and (**b**) TEM images of the AmG layer. Inset—corresponding ED pattern. (**c**–**e**) SEM images of the Am-ZnO flakes at different magnifications. (**f**,**g**) TEM images of the Am-ZnO layer at different magnifications. (**h**,**i**) HR-TEM images of the individual ZnO NPs, allowing for determination of the facet type. (**j**) ED pattern acquired for the area of the Am-ZnO displayed in (**f**).

**Figure 2 nanomaterials-14-00735-f002:**
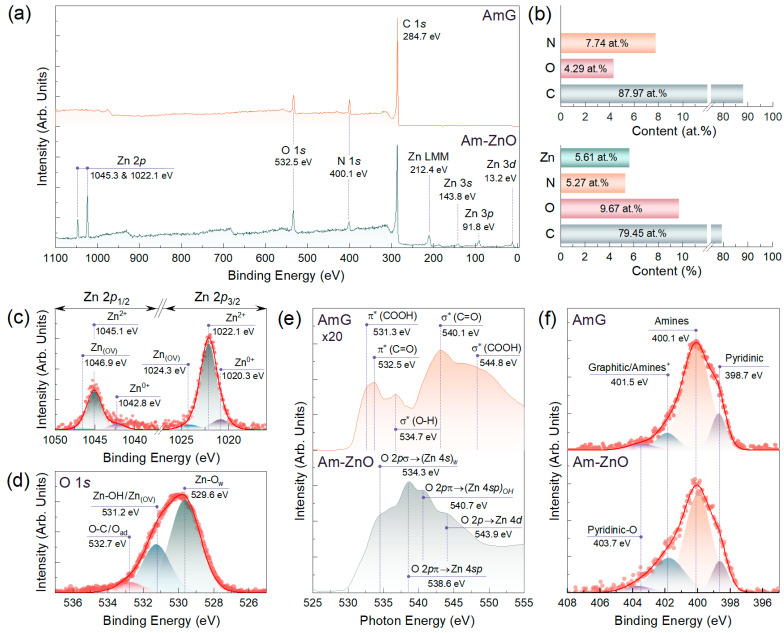
AmG and Am-ZnO chemistry examination by means of core-level methods: (**a**) Survey spectra and (**b**) bar chart displaying the elemental composition of AmG and Am-ZnO layers. (**c**) Zn 2*p* and (**d**) O 1*s* high-resolution XPS spectra of the Am-ZnO. (**e**) O *K*-edge XAS and (**f**) N 1*s* high-resolution XPS spectra of the AmG and Am-ZnO samples.

**Figure 3 nanomaterials-14-00735-f003:**
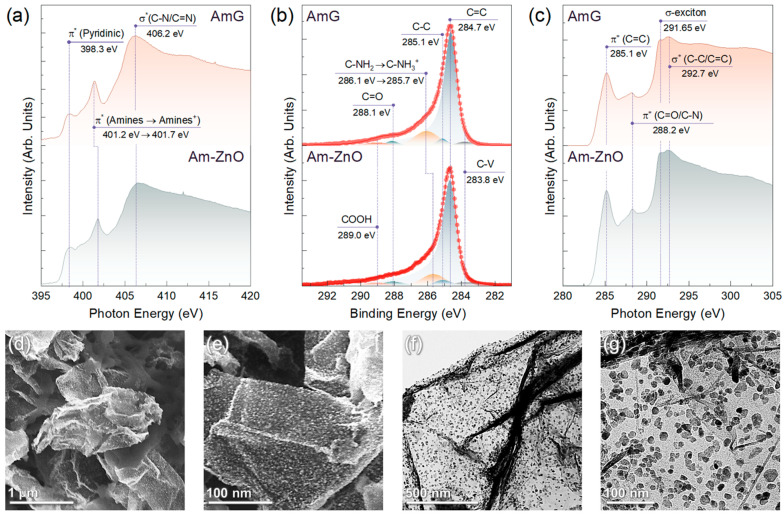
Spectroscopic and electron microscopy studies on the bonding of ZnO to AmG: (**a**) N *K*-edge XAS, (**b**) C 1*s* XPS, and (**c**) C *K*-edge XAS spectra of AmG and Am-ZnO layers. (**d**,**e**) SEM and (**f**,**g**) TEM images of the Am-ZnO layers after annealing at T = 315 °C.

**Figure 4 nanomaterials-14-00735-f004:**
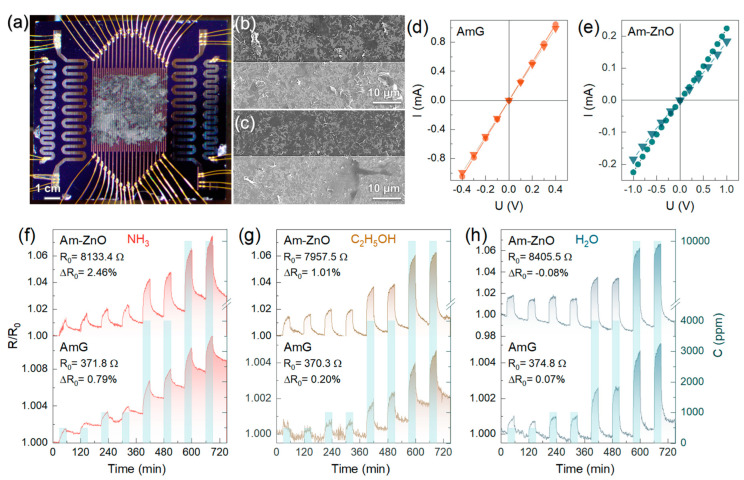
The gas-sensing performance of multisensor chips composed of AmG and Am-ZnO layers: (**a**) Optical photo of the AmG chip. (**b**,**c**) SEM images of the (**b**) AmG and (**c**) Am-ZnO layers covering the multi-electrode chip in the thinnest areas. (**d**,**e**) Forward and backward scans of the *I*-*V* curves recorded for (**d**) AmG and (**e**) Am-ZnO. (**f**–**h**) The resistance transient of the exemplary sensors composed of AmG and Am-ZnO upon exposure to (**f**) NH_3_, (**g**) C_2_H_5_OH, and (**h**) H_2_O in mixtures with dry air. The analyte concentrations are indicated by columns related to the second log-scaled ordinate axis, in the range of 500–10,000 ppm.

**Figure 5 nanomaterials-14-00735-f005:**
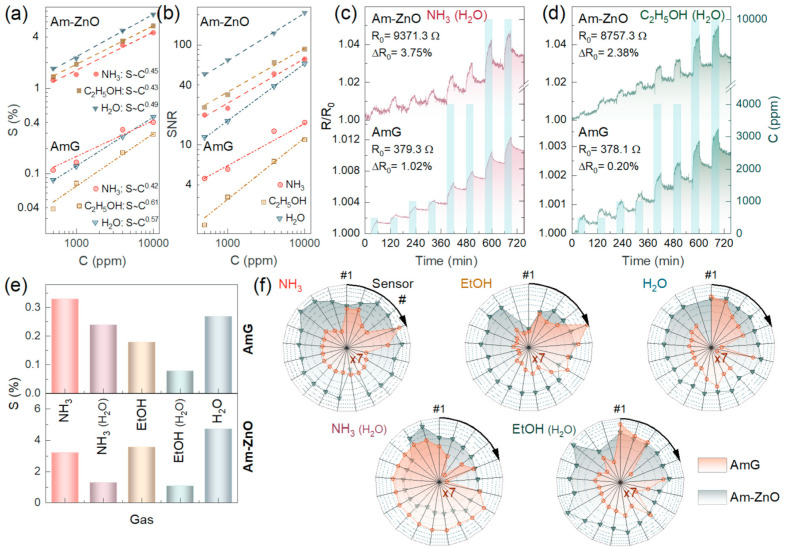
The gas-sensing performance of multisensor chips composed of AmG and Am-ZnO layers: (**a**) Chemiresistive response and (**b**) *SNR* magnitude of the exemplary sensor elements versus the concentrations of NH_3_, C_2_H_5_OH, and H_2_O; dash–dot lines refer to AmG, while dashed lines correspond to Am-ZnO. (**c**,**d**) The resistance transients of the exemplary sensors exposed to (**c**) NH_3_ and (**d**) C_2_H_5_OH in humid air, 25 rel.%. (**e**) The median chemiresistive response of the multisensor arrays to the test analytes in the both dry and humid air. (**f**) The normalized raw vector signals, as radar patterns, generated by the sensor arrays to all of the studied analytes in dry and humid air; AmG is orange, while Am-ZnO is dark cyan. S values for the AmG layer are additionally multiplied by a factor of 7 for explicit comparison of the AmG and Am-ZnO patterns in the radar diagram.

**Figure 6 nanomaterials-14-00735-f006:**
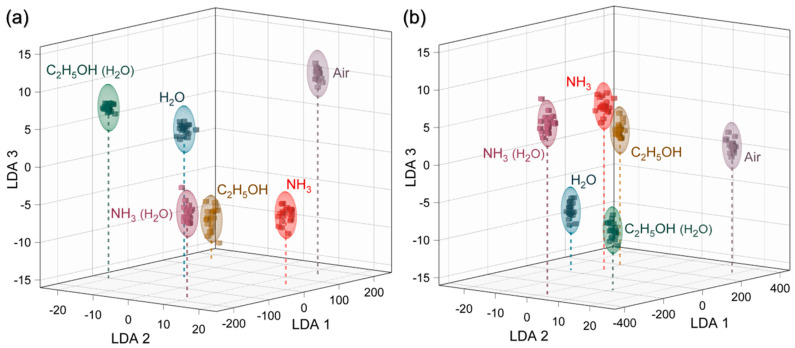
Recognition of the test analyte VOCs (4000 ppm concentration) by the LDA technique: (**a**,**b**) Primary 3D cross-section of 5D LDA total space for the data recorded for the whole set of analytes in dry air and humid air for the multisensor chips composed of (**a**) AmG and (**b**) Am-ZnO layers.

**Figure 7 nanomaterials-14-00735-f007:**
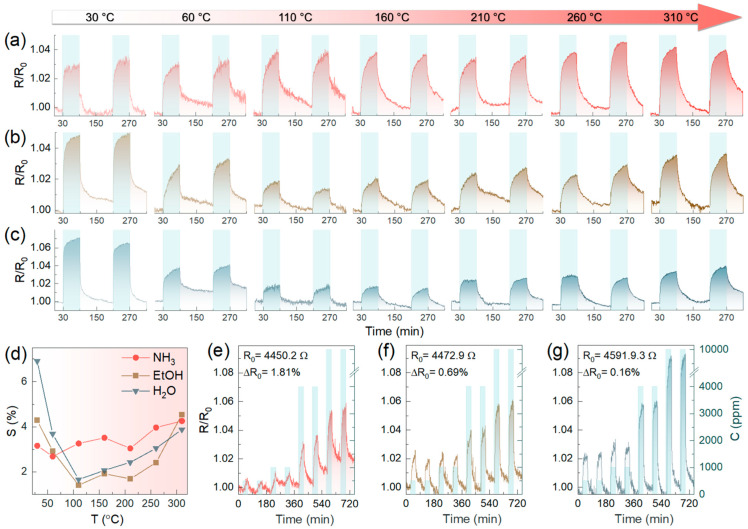
The temperature effect in the 30–310 °C range on the chemiresistive effect of the Am-ZnO-based multisensor chip: (**a**–**c**) The *R*(*t*) transients for the typical sensor elements of the chip under various operating temperatures while being exposed to (**a**) NH_3_, (**b**) C_2_H_5_OH, and (**c**) H_2_O. (**d**) The response value, *S*, dependent on the operating temperature. (**e**,**f**) The *R*(*t*) transients recorded while exposing the Am-ZnO chip to (**e**) NH_3_, (**f**) C_2_H_5_OH, and (**g**) H_2_O after high-temperature treatment.

## Data Availability

The data presented in this study are available upon request from the first author.

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
