# Peer review of "Rationalizing Graphene–ZnO Composites for Gas Sensing via Functionalization with Amines"

_nanomaterials, 2024, doi:10.3390/nano14090735_

Round 1
Reviewer 1 Report
Comments and Suggestions for Authors
In this article MK Rabchinskii et al reported the rational design of the graphene-metal oxide composites based on ZnO by applying graphene amination. The concept theme of the research work is new and novel and the manuscript design and writing is very good. Authors have done extensive research analysis and the material characterization analysis were extraordinarily explained well discrete level of analysis. Authors have additionally provided all the detailed information about the materials synthesis, fabrication details, characterization condition and sensing instrumentation very clearly which definitely useful for the aspirant researchers to gain much insightful knowledge for their new starts. The as prepared sensors showed significant enhancement of gas sensing properties of AmG at room temperature upon introduction of ZnO nano particles due to the formation of a network of p-n heterojunctions. Moreover, the employing the pattern recognition technique LDA analysis, gives full insights and clear understanding of the sensor’s identification in sensing. However, there are some minor issues that authors needs to clear before the final acceptance.
1) Please describe more in details about the aminated graphene, which is easy to understand for the readers.
2) How about the dangling bond present in graphene oxide/Am material which also acts a gas trapping centers for gas molecules.
3) Why does the author choose the ZnO nanostructure only for this research? Please emphasize the novelty of your structure. Also, please make a statement of choosing ZnO nanoparticle in your research as compared to another nanostructure. Introduction: (1) Why study ZnO based nanoparticles their wide application in various filed and how does it helpful to enhance gas sensing properties. How did the authors confess the strong correlation bestrewn the Am-ZnO p-n heterojunction interface and how to avoid the agglomeration? Some of the latest review reports are suggested to cite at the introduction discussions, ACS Materials Letters 5 (2023): 2739-2746. Materials Today Sustainability 25, 100649
4) In the pictographical representation please check that 0.5L/min (500 sccm) which was mistakenly added as 0,5L/min, alos check the entire image for errors in Figure S1.
5) What may be purity of the dry air.
6) It could be better if authors can present the gas concentration ppm level in the Figure 4 f-h which is easy for the readers to understand.
7) Please elaborate the basic concept of understanding the raw vector signals.
8) The gas concentration levels were still higher than the reported literatures, in which the gas sensors detection at lower temperatures and low detection ppm/ppb levels are having huge demand in the market. Please explain.
9) Which source have the authors used to perform the LDA analysis, MATLAB or Python ML tool. Please specify the tools in the revised manuscript and please try top add the data frame set and CSV files of your compromised data to read the files in the particular modules to run. Which also a provides new thoughtful insights to the researchers and readers to explore new data sets based on your results.
Reviewer 2 Report
Comments and Suggestions for Authors
A sensor array employs a rapid detection approach that imitates the olfactory system, relying on pattern recognition from the collective input of multiple sensor points rather than chemical specificity from a single sensor. This work reports on design of of the graphene-metal oxide composites for gas sensing. The graphene was aminated and the amine function was used for subsequent grafting of ZnO nanoparticles (NPs). Then the functionalized graphene with ZnO NPs was characterized by different technique including spectroscopy methods, revealing the formation of a network of ionic bonds between the aminated graphene (AmG) and ZnO. On-chip sensor arrays comprised of both pristine AmG and ZnO functionalized Am were fabricated and thoroughly tested, demonstrating enhancement of gas sensing properties of AmG at room temperature upon introduction of ZnO NPs due to the formation of an heterostructure between graphene/ZnO. Employing the pattern recognition technique, namely linear discriminant analysis, to process the acquired multisensor response, precise identification of the studied analytes, ammonia, ethanol and water in both dry and air has been realized. Overall the work contains useful research, a lot of interesting results, a huge quantity of results and scientifically sounds good but a revision is required. The quality of English language used in this manuscript is very poor. I would suggest to the authors to use a professional English editing service to improve their manuscript. Here are my concerns.
1) Line 45, the common abbreviation of metal oxide is MO not MeO
2)Line 22 " ZnO nanoparticles across the graphene" can be rewritten as " ZnO nanoparticles through the graphene"
3) Line 25 The stability of the acquired nanocomposite (Am-ZnO) has been justified by revealing no noticeable changes in its morphology upon its long-term heating up to 350 °C employing electron microscopy. can be rewritten as " The stability of the resulting Am-ZnO nanocomposite has been confirmed by demonstrating that its morphology remains unchanged even after prolonged heating up to 350°C, as observed by electron microscopy.
4) Line 67 " however etc..." The meaning of this this sentence is not immediately clear. Please reformulate it.
5) Line 72 " manageable work function" can be changed to " tunable work function"
6) Line 158 Please reformulate the sentence as "The Core-level spectroscopy was used to study the chemical composition of materials under study was, namely X-ray photoelectron spectroscopy (XPS) and X-ray absorption spectroscopy (XAS), which carried out at the ultra-high-vacuum experimental station of the Russian German beamline of electron storage ring BESSY-II at Helmholtz-Zentrum Berlin (HZB)"
Line 292 " are observed amid the folds," please rewrite as " are observed amidst the folds"
Line 312 sub-section title as "Chemistry of the AmG and Am-ZnO" is not clear and ambiguous
Line 395 the authors stated that " This implies no covalent bonding between ZnO nanoparticles and amine groups through the following possible reaction". However the zinc has a tendency to form covalent bond with nitrogen atom of the amine group due to the basic character of amine function and acid character of zinc ion enabling strong formation of covalent bond. Please comment this.
Comments on the Quality of English Language.I would recommend to the authors to use a professional English editing service to improve their manuscript
Reviewer 3 Report
Comments and Suggestions for Authors
In this paper (nanomaterials-2937292), the authors exhibited multisensor arrays on-chip based on aminated graphene (AmG)-ZnO for gas detections at room temperature. The strategy is interesting, and the data results are complete. But before accepting, some issues need to be addressed as follows.
1. Abstract: It is recommended to provide some key gas sensing test results.
2. Introduction: The problems faced by gas sensors and the problems attempted to be addressed in this study were not clearly explained.
3. According to the response definition (equation (1)), does it mean that the resistance of the sensor to all test gases increases? What about humidity? NO2 (although NO2 was not tested)?
4. The definition of sensor response (equation (1)) and its representation in the Figure 4, 5 and 7 (R/R0) are inconsistent.
5. Figure 2d: For O1s, it can be divided into lattice oxygen, oxygen vacancies, and adsorbed oxygen, which is more conducive to establishing a connection with gas sensing characteristics. Chemosensors 2024, 12(3), 43 can be referenced and cited.
6. “…several analytes, particularly ammonia (NH3), ethanol (C2H5OH), and water (H2O) vapors”. What is the motivation and basis for selecting test subjects?
7. Figure 4d, e : “…linear I-V curves are observed…”. This also indicates that excluding the contribution of electrodes to gas sensing (such as non-Schottky contact), the gas sensing characteristics originate from the gas sensing material itself. Suggest providing additional explanations.
8. To evaluate the gas sensing performances, it is recommended to compare it with relevant reports.
9. Please explain the gas sensing mechanism. Especially regarding the response mechanism to humidity (water molecules).
10. The numbers in the chemical formula require subscripts, including references.
11. Reference list: Gas sensors have developed rapidly, and it is recommended to focus on the literature from the past three years.
Comments on the Quality of English LanguageMinor editing of English language required.
Round 2
Reviewer 3 Report
Comments and Suggestions for Authors
The response and revised manuscript are satisfactory, and it is recommended to accept.